# Contraceptive-Pill-Sourced Synthetic Estrogen and Progestogen in Water Causes Decrease in GSI and HSI and Alters Blood Glucose Levels in Climbing Perch (*Anabas testudineus*)

Chathuri Weerasinghe [1,2,†], Noreen Akhtar [3,†], Md Helal Uddin [4,5,†], Mahesh Rachamalla [5], Kizar Ahmed Sumon [4], Md. Jakiul Islam [6], Ramji Kumar Bhandari [7] and Harunur Rashid [4,*]

1 Environmental Sciences Program, Asian University for Women, Chittagong 4000, Bangladesh
2 Department of General Education, BRAC University, Dhaka 1212, Bangladesh
3 Department of Sustainability and Compliance, The All Pakistan Textile Mills Association, Islamabad 44000, Pakistan
4 Department of Fisheries Management, Bangladesh Agricultural University, Mymensingh 2202, Bangladesh
5 Department of Biology, University of Saskatchewan, Saskatoon, SK S7N 5E2, Canada
6 Faculty of Fisheries, Sylhet Agricultural University, Sylhet 3100, Bangladesh
7 Department of Biology, University of North Carolina at Greensboro, Greensboro, NC 27412, USA
* Correspondence: rashid@bau.edu.bd
† These authors contributed equally to this work.

**Abstract:** The present study was conducted to understand the changes in gonads and hematological parameters in climbing perch (*Anabas testudineus*) exposed to synthetic estrogen and progestogen [mixture of ethinylestradiol (EE2) and desogestrel (DES)]. Climbing perch were exposed to four different concentrations of EE2/DES mixtures, viz. 0 ng of EE2 and DES/L (T0), 3 ng EE2 and 15 ng DES/L (T3), 30 ng EE2 and 150 ng DES/L (T30), and 300 ng EE2 and 1500 ng DES/L (T300) for 60 days. On days 45 and 60, samples were taken to assess changes in somatic indexes, gonad histology, and hematological parameters. The gonadosomatic index (GSI) increased in both females and males with increasing concentrations of estrogen mixtures except for T30 females, which was the lowest among all\four treatments. The hepatosomatic index (HSI) was observed to be increased in males as estrogen content increased. However, compared to fish at T0, HSI in female individuals did not vary in T30 fish, where the value was the highest among all the treatments. On day 45, histological observations showed no feminization or intersexuality but several germ-cell deformities in the ovary (adhesion, degenerated oocyte wall, degenerated granulose layer, increased interfollicular space, atretic follicle, and cytoplasmic clumping) and testes (increased interstitial area, focal loss of spermatocyte, dilation of the lumen, breakage of tubular epithelium, and elongated seminiferous tubule) were observed in fish exposed to EE2 and DES. Fish reared at T30 had lower RBC count, hemoglobin (Hb), glucose, and hematocrit levels. On day 60, fish reared at T30 had the highest Hb content compared to fish raised in other treatment conditions. WBC was progressively higher with increasing EE2/DES concentrations. Significant erythrocyte cytoplasmic abnormalities and erythrocyte nuclear abnormalities were observed in fish exposed to higher EE2/DES concentrations. The present study provides insights into the adverse impacts of synthetic estrogens sourced from human contraceptive pills on fish physiology.

**Keywords:** synthetic estrogen; climbing perch; *Anabas testudineus*; gonad; hematology; deformity

## 1. Introduction

Pharmaceuticals are a broad category of medicinal compounds that are used to diagnose, treat, mitigate, or prevent diseases. A wide range of human and veterinary pharmaceuticals pollutes both freshwater and marine ecosystems [1]. Several scientific studies have reported that a group of active pharmaceutical ingredients (APIs) were found in

surface water, groundwater, effluents, and sediments, and even in biota [2–4]. The APIs in aquatic ecosystems find a route to the water from pharmaceutical manufacturing plants, municipalities, hospitals, industrial wastewater treatment plants, livestock farming, and aquaculture, among other sources [5–9]. After entering aquatic environments, APIs might have negative impacts on aquatic organisms including fish [10]. APIs can affect a variety of physiological processes, including xenobiotic biotransformation, reactive oxygen species control, hormone synthesis, and reproduction of aquatic organisms [11–14].

Ethinylestradiol and desogestrel pills (commonly called Desogen pills or ipills) are widely used oral contraceptive pills (OCP) in many parts of the world including Bangladesh. OCPs have been highlighted as one of the most important problems in the context of pharmaceutical toxicity in aquatic environments due to the endocrine disruptive qualities they possess, as consumption rates have grown dramatically over the previous decades [15,16]. Globally, around 8% of reproductively capable women use OCPs and for Bangladesh, it is 23.1%, which is the highest in Asia. A small country such as Bangladesh, having a high population (>166 million and the tenth most densely populated country in the world) with a wide riverine network, receives anthropogenically originated wastes. Thus, OCP residues easily find their way into water bodies [17–19]. Along with growing levels of environmental pollutants [20–22], emerging threats from pharmaceutical contaminants [23–26] will be a great threat to aquatic species' wellbeing.

Desogen pills contain 0.15 mg desogestrel (DES) and 0.03 mg ethinylestradiol (EE2) [27]. As a synthetic estrogen, EE2's main effects are on the sex hormones of animals. It can cause alterations in fish gonads during the gonadal differentiation phase, alter testosterone/estrogen levels, and affect osmoregulation, growth, reproduction, and development [28]. In a previous study, chronic exposure of fathead minnow (*Pimephales promelas*) to sub-lethal concentrations of the estrogenic pharmaceutical EE2 showed a negative effect on the sustainability of wild fish populations [29]. Similar to EE2 and other synthetic estrogens, synthetic progesterone or progestin have also been identified as endocrine disruptors due to their ability to hinder male reproduction and alter testicular differentiation and development [30].

The toxicity of emerging pharmaceutical contaminants on non-target aquatic animals can be measured by observing physiological, biochemical, or molecular changes in fish species under laboratory conditions. This study can serve as an early warning for potential environmental pollution [31,32]. These early warning biomarkers can be used to detect the interaction of xenobiotics in the environment. Histopathological changes in fishes' internal organs are widely used biomarkers to detect the impacts of pollutants (metals, metalloids, persistent organic pollutants) on fish physiology [33–35]. Hematological parameters are also commonly used parameters to understand the health status of fish and the aquatic environment [36,37]. Blood parameters can reveal a lot about fish's physiological response to stressful environments [38–42]. Analyzing hematological and biochemical parameters in fish can help in determining animal health, welfare status, and habitat conditions as well [43–45]. Alterations in blood parameters associated with environmental pollutants represent an important tool in order to assessing the health of fish [46,47].

Climbing perch (*A. testudineus*) is a freshwater fish species that quickly adapts to and lives in the waters of rice fields, ponds, lakes, lagoons, and rivers, and is especially adaptable to living in dirty water. Climbing perch is a carnivorous, air-breathing, small indigenous fish found in shallow freshwater habitats and floodplain areas (paddy fields, ditches, and streams with dense vegetation) in Asia [48–50]. This species is frequently employed in aquaculture in South Asian countries, generating considerable earnings for farmers, and is particularly popular since it is resilient, long-lived, and tolerant to a high stocking rate. This species is found in Bangladesh, India, the Indochina Peninsula, southern China, Taiwan, the Philippines, and Indonesia [51]. Laboratory simulation tests could be an efficient research model for evaluating the effects contraceptive-pill-supplied estrogen effects on *A. testudineus* in nature [52–56]. The levels of hemoglobin (Hb), glucose, hematocrit, and RBC count in the blood of *A. testudineus* have been extensively utilized

as stress indicators and have been proven to vary in the presence of pollutants [1,57,58]. As a result, the purpose of this research is to assess the endocrine stress response. The current study sheds light on the negative effects of synthetic estrogens derived from human contraceptive tablets on fish physiology.

## 2. Materials and Methods

### 2.1. Experimental Fish

Healthy and active climbing perch (total length = 13.85 ± 0.60 cm; body weight = 95.56 ± 0.78 g) were procured from a local fish farm. A total of 180 *A. testudineus* adult fish was used (120 female, average length 14.85 cm, average weight 111.4 g; 60 male, average length 13.2 cm, average weight 78.1 g). Fish were kept in 10 L polyethylene bags filled with oxygenated water and were transported to the Asian University for Women (AUW), Chattogram, Bangladesh. To prevent possible infection, fish were washed with a 0.1% $KMnO_4$ solution before stocking at the AUW laboratory. Male and female fish were distinguished by observing the external genital opening and by stroking the smoothness of the operculum. A commercial feed was fed at 5% per kg body weight per day in two daily rations (Popular Poultry & Fish Feeds Ltd., Dhaka, Bangladesh, 32% protein). Fish handling and experimental protocols were approved by the Institutional Review Board of the Asian University for Women.

### 2.2. Experimental Setup and Chemical Exposure

Mature climbing perch were selected for the experiment and were kept in twelve previously prepared PVC tanks containing 100 L dechlorinated tap water in each tank equipped with sufficient aeration. Fifteen male and fifteen female fish were stocked in each tank and acclimatized to laboratory conditions for 21 days prior to starting exposure to four different concentrations (in triplicates) of EE2/DES mixture, viz., 0 ng EE2 plus 0 ng DES/L of water (T0), 3 ng EE2 plus 15 ng DES/L (T3), 30 ng EE2 plus 150 ng DES/L (T30), and 300 ng EE2 plus 1500 ng DES/L (T300). Marvelon tablets (ipill) (each pillcontaining 0.15 mg of desogestrel and 0.03 mg of ethinylestradiol), manufactured by Nuvista Pharma Ltd. (Nuvista Pharma 2021, Dhaka, Bangladesh), were purchased, ground, and mixed well in water to prepare the desired concentrations of EE2/DES. The desired levels of EE2/DES were selected based on environmentally detected concentrations reported in studies [59,60]. Dissolved oxygen as a water quality parameter was monitored daily and temperature was controlled at 27 °C; measurements were taken to maintain standard levels. No fish mortality was found during the acclimatization period. Fish were fed twice a day up to satiation and photoperiod was maintained at 12/12 h light/dark.

### 2.3. Collection of Gonads and Liver and Estimation of Gonadosomatic Index (GSI) and Hepatosomatic Index (HSI)

Fish were sampled on days 45 and 60. After collection, fish were immediately euthanized with a lethal dosage of MS-222 (500 mg $L^{-1}$). On each sampling day, three fish were sampled per replicate tank. Before dissection, fish length and weight were recorded. After dissection, gonad and liver weights were recorded using an electronic balance. Gonad samples were then rinsed with physiological saline solution and transferred to 10% buffered formalin solution at ambient temperature for tissue fixation. The GSI and HSI were measured using the following formulae [61].

$$GSI = [(gonad\ weight/body\ weight) \times 100]$$

$$HSI = [(liver\ weight/body\ weight) \times 100]$$

### 2.4. Histopathological Observations of Gonads

Histopathological analysis of the fixed gonad samples was conducted at the Department of Fisheries Management, Bangladesh Agriculture University, Mymensingh-2202, Bangladesh. Histopathology protocols were performed according to the methods described

in Sumon et al. (2019) [62]. Briefly, the fixed gonad tissues were first washed with running tap water and dehydrated in a graded series of ethyl alcohol, followed by clearing in benzene and embedding in paraffin. The embedded tissues were then sectioned using a microtome (5 μm thickness) followed by staining in hematoxylin–eosin. This way, six slides were prepared from a single tissue block. Finally, histopathological alterations were recorded and photographed using a photomicroscope (Olympus CX 41). Recorded histopathological alterations were then semi-quantified according to Sumon et al., 2019 [62].

*2.5. Measurement of Hemato-Biochemical Parameters*

Immediately after euthanasia, fish blood samples were taken from the caudal vein using a sterile plastic syringe. Blood samples immediately analyzed with 5 μL blood added to 995 μL RBC fluid, 5 μL blood added to 195 μL WBC fluid, and counts taken using a Neubauer chamber. For glucose testing, a glucose meter and strips were used. Blood was smeared and stained with Geimsa's stain then dried for microscopic observation. A proportion of this blood was pushed into a sterilized centrifuge tube containing anticoagulant (20 mM EDTA). The blood collection process for each fish was performed within a minute to minimize handling stress impacts on normal blood values. Blood glucose (mg/dl) was measured using glucose strips (Model: ET-232, Bioptik Technology Inc., Taipei, Taiwan 35057) fitted in a digital dual-function monitoring device (EasyMate®GHb). Hemoglobin (Hb %) was measured using SAHLI's hemometer (Model-3243000, MARIENFELD, Berlin, Germany). Hematocrit (%), PCV, MCV, MCH, and MCHC were calculated accordingly [63]. Total counts of RBCs and WBCs were conducted using the established Neubauer hemocytometer counting method.

*2.6. Erythrocytic Cellular Abnormalities (ECA)*

Just after blood collection, blood was smeared on clean microscope slides and air-dried for 10 min. The slides were washed with 100% alcohol and dried for a further 10 min. They were then stained with 5% Giemsa stain for 12–15 min followed by rinsing in distilled water for 4–5 min. Then the slides were air-dried overnight followed by mounting with DPX. After that ECAs were observed under a $100\times$ optical microscope. Six slides were prepared from the blood of each sampled fish and a total of 2000 cells was scored from those slides. Only cells having intact cellular and nuclear membranes were scored. ECAs were classified according to Carrasco et al., 2011 [64].

*2.7. Statistical Analysis*

Values were presented as mean ± standard deviation (SD). A one-way ANOVA was used to compare the means of treatments, followed by Tukey's post-hoc multiple comparison test. Statistical significance was set at $p \leq 0.05$. Data were analyzed using SPSS version 14.0.

**3. Results**

*3.1. Changes in GSI and HSI*

GSI in male climbing perch was significantly higher in T3 and T30 compared to T300 and control (T0) on days 45 and 60. For fish in all treatment conditions, the GSI value for female individuals was significantly higher compared to fish in the control treatment (Table 1). Throughout the study, compared to the control treatment, in fish exposed to estrogen mixtures, HSI was higher for male individuals. In contrast, HSI values for female individuals did not differ significantly (Table 1). However, for both GSI and HSI values, fish reared under T30 conditions had the highest values.

**Table 1.** Changes in GSI and HSI of climbing perch exposed to different concentrations of ethinylestradiol/desogestrel mixtures.

| Parameters | Treatments | Sampling Days | | | |
|---|---|---|---|---|---|
| | | Male | | Female | |
| | | Day 45 | Day 60 | Day 45 | Day 60 |
| GSI (± S.D.) | T0 | 0.42 ± 0.12 [a] | 0.62 ± 0.35 [a] | 3.57 ± 0.12 [a] | 5.92 ± 1.59 [a] |
| | T3 | 0.68 ± 0.09 [b] | 1.12 ± 0.29 [c] | 6.34 ± 0.15 [b] | 7.64 ± 0.45 [b] |
| | T30 | 0.74 ± 0.15 [b] | 0.88 ± 0.26 [b] | 7.78 ± 0.11 [b] | 9.54 ± 0.92 [b] |
| | T300 | 0.57 ± 0.11 [a] | 0.67 ± 0.34 [a] | 9.03 ± 0.19 [c] | 13.33 ± 0.44 [c] |
| HSI (± S.D.) | T0 | 1.04 ± 0.53 [a] | 1.15 ± 0.59 [a] | 1.28 ± 0.08 [a] | 1.34 ± 0.04 [b] |
| | T3 | 1.48 ± 0.11 [b] | 1.54 ± 0.27 [b] | 1.22 ± 0.12 [a] | 1.03 ± 0.09 [a] |
| | T30 | 1.73 ± 0.36 [c] | 1.74 ± 0.23 [c] | 1.19 ± 0.06 [a] | 1.19 ± 0.24 [a] |
| | T300 | 1.53 ± 0.07 [b] | 1.49 ± 0.29 [b] | 1.12 ± 0.01 [a] | 1.11 ± 0.40 [a] |

T0, T3, T30, and T300 correspond to 0 ng EE2 plus 0 ng DES, 3 ng EE2 plus 15 ng DES, 30 ng EE2 plus 150 ng DES, and 300 ng EE2 plus 1500 ng DES per liter of water fishes were exposed to. Values with different alphabetical superscripts in a row differ significantly ($p < 0.05$) among different treatments. All values were expressed as mean ± SD.

### 3.2. Changes in Ovarian Histology

Ovaries from controls (T0) did not show any histopathological alterations during the experimental period. Ovaries of fish in the control treatment (T0) had normal nuclei which were characterized by round-shaped, regularly structured yolk vesicles and granulose layers (Figure 1A,B). However, estrogen dosage- and time-dependent histopathological alterations in the ovary were evident (Figure 1C–H). After 45 and 60 days of exposure, histopathological alterations such as degeneration of the granulose layer (DGL), degeneration of the oocyte wall (DOW), and atretic follicles (AF) were observed in T3 (Figure 1C,D); interfollicular spaces (IFS) and cytoplasmic clumping (CC) were observed in T30 (Figure 1E,F); and atretic follicles (AF), necrosis (NE), and cysts (CT) were observed in T300 (Figure 1G,H). At this stage, T300 ovaries were found to have severe DGL and IFS deformities. On the other hand, histological alterations similar to those above were observed in all the treatments marked with an increase in the intensity of deformities (Table 2).

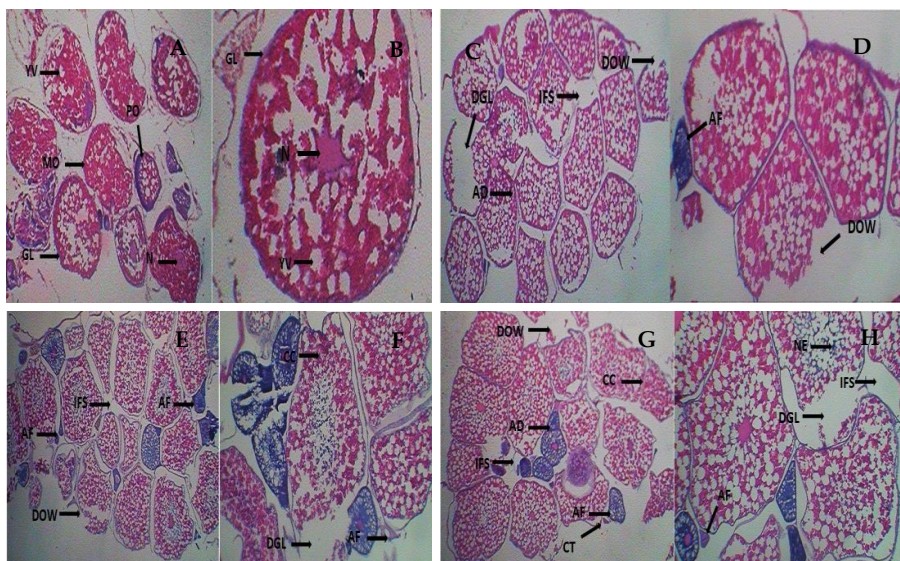

**Figure 1.** Histological alteration in climbing perch ovaries exposed to different concentrations of ethinylestradiol (EE2)/desogestrel (DES) mixture. (**A–H**) correspond to 0 ng EE2 plus 0 ng DES, 3 ng

EE2 plus 15 ng DES, 30 ng EE2 plus 150 ng DES, and 300 ng EE2 plus 1500 ng DES per liter of water fishes were exposed to, respectively; arrows indicate mature oocyte (MO), pre-mature oocyte (PO), nucleus (N), yolk vesicle (YV), granulose layer (GL), degenerated oocyte wall (DOW), wrinkled oocyte (WO), atretic follicle (AF), degenerated granulose layer (DGL), interfollicular space (IFS), adhesion (AD), cytoplasmic clumping (CC), cysts (CT), disrupted oocytes (DO), and necrosis (NE); H–E staining; (**A,C,E,G**) are 4× and (**B,D,F,H**) are 10× magnified.

**Table 2.** Summary of histopathological alterations of climbing perch ovaries exposed to different concentrations of ethinylestradiol/desogestrel mixtures.

| Alterations | Treatment | Exposure Duration (Days) | |
|---|---|---|---|
| | | 45 | 60 |
| Degenerated granulose layer (DGL) | T0 | − | − |
| | T3 | − | + |
| | T30 | + | ++ |
| | T300 | +++ | +++ |
| Atretic follicle (AF) | T0 | − | − |
| | T3 | + | + |
| | T30 | − | ++ |
| | T300 | ++ | +++ |
| Inter follicular space (IFS) | T0 | − | − |
| | T3 | + | + |
| | T30 | ++ | + |
| | T300 | +++ | +++ |
| Adhesion (AD) | T0 | − | − |
| | T3 | − | + |
| | T30 | + | ++ |
| | T300 | ++ | +++ |
| Cytoplasmic clumping (CC) | T0 | − | − |
| | T3 | − | − |
| | T30 | + | ++ |
| | T300 | + | ++ |
| Cysts (CT) | T0 | − | − |
| | T3 | − | + |
| | T30 | ++ | + |
| | T300 | ++ | ++ |
| Degenerated oocyte wall (DOW) | T0 | − | − |
| | T3 | + | + |
| | T30 | ++ | ++ |
| | T300 | ++ | ++ |
| Necrosis (NE) | T0 | − | − |
| | T3 | − | + |
| | T30 | + | + |
| | T300 | ++ | ++ |

T0, T3, T30, and T300 correspond to 0 ng EE2 plus 0 ng DES, 3 ng EE2 plus 15 ng DES, 30 ng EE2 plus 150 ng DES, and 300 ng EE2 plus 1500 ng DES per liter of water fishes were exposed to, respectively. −, none (0%); +, mild (<10%); ++, moderate (10 to 50%); +++, severe (>50%).

### 3.3. Changes in Histo-Architecture of Testes

Testes from the control group (T0) did not show any histopathological alterations during the experimental period. Regularly shaped seminiferous tubules (ST) bordered with interstitial connective tissues (IST) were filled and packed with numerous spermatozoa (SZ) (Figure 2A,B). For both sampling days, EE2/DES treatments caused various dosage- and time-dependent histopathological alterations in the testes (Figure 2C–H). Testis sections from estrogen-treated fishes showed several histopathological alterations such as ruptured interstitial connective tissue (rIST), deformed seminiferous tubules (dST), empty tubular lumen (eL), and loss of spermatozoa (sSZ). Concerning the intensity of deformities, there were no major differences between 45-day and 60-day exposures. However, the intensity of deformities increased with increasing concentrations of EE2/DES. Two deformities, eL and sSZ, were recorded as 'severe' in T300 fish on both sampling days (Table 3).

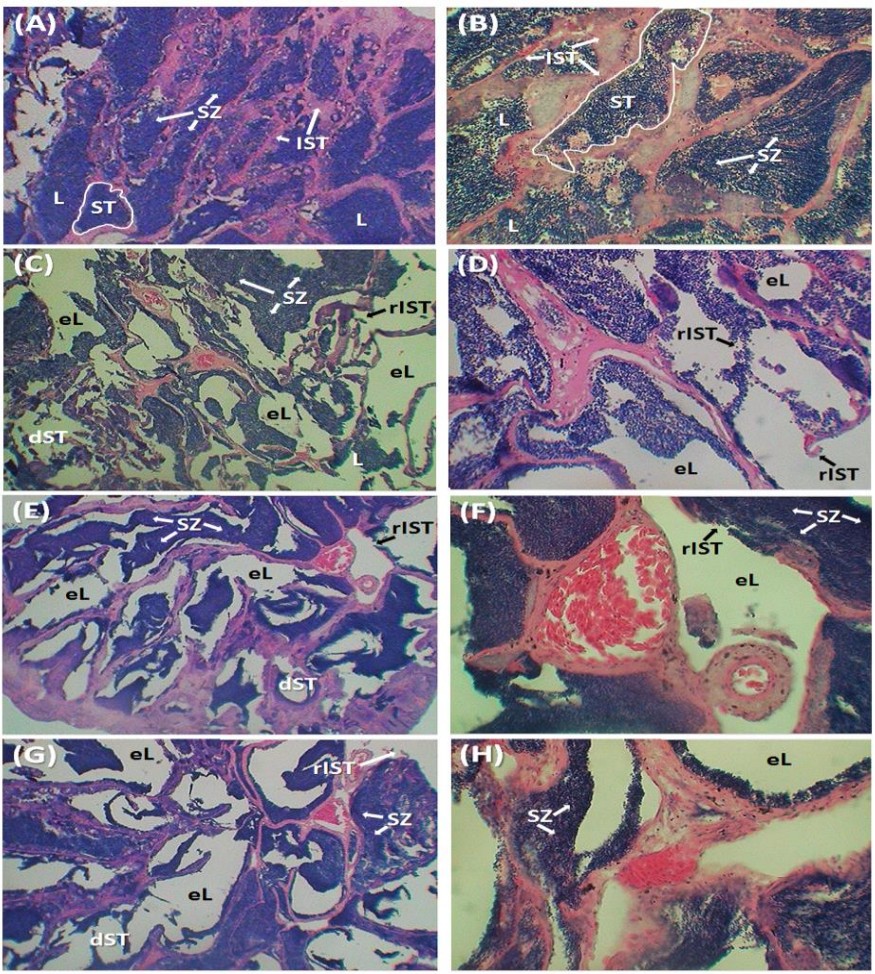

**Figure 2.** Histological alterations in climbing perch testes exposed to different concentrations of ethinylestradiol (EE2)/desogestrel (DES) mixture. (**A–H**) correspond to 0 ng EE2 plus 0 ng DES, 3 ng EE2 plus 15 ng DES, 30 ng EE2 plus 150 ng DES, and 300 ng EE2 plus 1500 ng DES per liter of water fishes were exposed to, respectively; arrows indicate seminiferous tubules (ST), Sertoli cells (S), increased interstitial area (IIA), broken testicular epithelium (BTE), absent Sertoli cells (ASC), degenerated Sertoli cells (DSC), cysts (CT), and focal loss of spermatocytes (FLS); H–E staining; (**A,C,E,G**) are 10× and (**B,D,F,H**) are 40× magnified.

**Table 3.** Summary of histopathological alterations of climbing perch testes exposed to different concentrations of ethinylestradiol/desogestrel mixtures.

| Alterations | Treatment | Exposure Duration (Days) | |
| --- | --- | --- | --- |
| | | 45 | 60 |
| Ruptured interstitial connective tissue (rIST) | T0 | − | − |
| | T3 | + | ++ |
| | T30 | ++ | ++ |
| | T300 | ++ | ++ |
| Deformed seminiferous tubule (dST) | T0 | − | − |
| | T3 | + | + |
| | T30 | + | + |
| | T300 | ++ | ++ |
| Empty tubular lumen (eL) | T0 | − | − |
| | T3 | ++ | ++ |
| | T30 | ++ | ++ |
| | T300 | +++ | +++ |
| Loss of spermatozoa (sSZ) | T0 | − | − |
| | T3 | + | ++ |
| | T30 | ++ | ++ |
| | T300 | +++ | +++ |

T0, T3, T30, and T300 correspond to 0 ng EE2 plus 0 ng DES, 3 ng EE2 plus 15 ng DES, 30 ng EE2 plus 150 ng DES, and 300 ng EE2 plus 1500 ng DES per liter of water fishes were exposed to, respectively. −, none (0%); +, mild (<10%); ++, moderate (10 to 50%); +++, severe (>50%).

### 3.4. Changes in Hemato-Biochemical Parameters

On both sampling days, blood glucose levels were suppressed significantly in males with increasing concentrations of EE2/DES mixture. Opposite to males, on day 45, blood glucose levels were elevated in females with increasing concentrations of EE2/DES mixtures. In general, for both male and female individuals, Hb (%) level was found to be decreased with increasing concentrations of EE2/DES. With increasing hormone concentrations, overall, RBC values were decreased opposite to WBC. Similar to RBC, hematocrit values were reduced reciprocally with higher concentrations of estrogen mixtures. MCV, MCH, and MCHC values did not differ significantly between treatments or sexes (Table 4).

**Table 4.** Changes in hemato-biochemical parameters of climbing perch exposed to different concentrations of ethinylestradiol/desogestrel mixtures.

| Parameters | Treatments | Sampling Days | | | |
| --- | --- | --- | --- | --- | --- |
| | | Male | | Female | |
| | | 45 Day | 60 Day | 45 Day | 60 Day |
| Blood glucose (mg/dL) | T0 | 168.3 ± 0.03 [c] | 181.3 ± 0.03 [c] | 86.0 ± 0.05 [a] | 120.3 ± 0.03 [b] |
| | T3 | 101.3 ± 0.04 [b] | 122.0 ± 0.07 [b] | 131.3 ± 0.07 [b] | 111.3 ± 0.05 [b] |
| | T30 | 117.6 ± 0.06 [b] | 110.3 ± 0.09 [a] | 120.3 ± 0.11 [b] | 104.3 ± 0.07 [a] |
| | T300 | 85.0 ± 0.07 [a] | 112.3 ± 0.11 [a] | 153.3 ± 0.01 [c] | 118.0 ± 0.03 [b] |
| Blood hemoglobin (Hb %) | T0 | 7.5 ± 0.01 [b] | 11.9 ± 0.03 [b] | 6.6 ± 0.01 [b] | 13.0 ± 0.01 [c] |
| | T3 | 8.0 ± 0.07 [b] | 11.6 ± 0.05 [b] | 6.9 ± 0.03 [b] | 12.8 ± 0.03 [b] |
| | T30 | 6.4 ± 0.11 [a] | 10.5 ± 0.19 [a] | 5.3 ± 0.05 [a] | 11.9 ± 0.05 [a] |
| | T300 | 6.1 ± 0.18 [a] | 12.3 ± 0.28 [c] | 5.6 ± 0.17 [a] | 11.6 ± 0.19 [a] |

**Table 4.** *Cont.*

| Parameters | Treatments | Sampling Days | | | |
|---|---|---|---|---|---|
| | | Male | | Female | |
| | | **45 Day** | **60 Day** | **45 Day** | **60 Day** |
| RBC ($\times 10^6/mm^3$) | T0 | 0.19 ± 0.01 [a] | 0.39 ± 0.01 [c] | 0.25 ± 0.01 [b] | 0.31 ± 0.01 [c] |
| | T3 | 0.31 ± 0.04 [b] | 0.33 ± 0.05 [b] | 0.25 ± 0.07 [b] | 0.24 ± 0.05 [b] |
| | T30 | 0.19 ± 0.04 [a] | 0.25 ± 0.04 [a] | 0.17 ± 0.04 [a] | 0.20 ± 0.04 [a] |
| | T300 | 0.18 ± 0.11 [a] | 0.27 ± 0.17 [a] | 0.18 ± 0.09 [a] | 0.17 ± 0.07 [a] |
| WBC ($\times 10^3/mm^3$) | T0 | 1.19 ± 0.03 [b] | 1.12 ± 0.03 a | 0.96 ± 0.05 [a] | 1.01 ± 0.07 [a] |
| | T3 | 1.01 ± 0.05 [a] | 1.37 ± 0.04 b | 0.88 ± 0.03 [a] | 1.13 ± 0.09 [b] |
| | T30 | 1.08 ± 0.04 [a] | 1.42 ± 0.05 b | 1.15 ± 0.04 [b] | 1.35 ± 0.11 [b] |
| | T300 | 1.47 ± 0.13 [c] | 1.55 ± 0.17 c | 1.29 ± 0.04 [c] | 1.50 ± 0.13 [c] |
| Hematocrit (%) | T0 | 22.5 ± 0.03 [b] | 35.7 ± 0.03 [b] | 19.8 ± 0.05 [b] | 39.0 ± 0.05 [c] |
| | T3 | 24.0 ± 0.05 [c] | 34.8 ± 0.09 [b] | 20.7 ± 0.07 [b] | 35.1 ± 0.05 [a] |
| | T30 | 19.2 ± 0.06 [a] | 31.5 ± 0.06 [a] | 15.9 ± 0.09 [a] | 35.7 ± 0.09 [a] |
| | T300 | 18.3 ± 0.09 [a] | 36.9 ± 0.11 [b] | 16.8 ± 0.07 [a] | 34.8 ± 0.09 [a] |
| Mean corpuscular volume (MCV) | T0 | 89.9 ± 0.05 | 90.0 ± 0.09 | 90.1 ± 0.03 | 89.8 ± 0.09 |
| | T3 | 90.0 ± 0.03 | 89.9 ± 0.03 | 90.1 ± 0.04 | 90.0 ± 0.03 |
| | T30 | 90.1 ± 0.06 | 90.2 ± 0.05 | 90.4 ± 0.17 | 89.9 ± 0.07 |
| | T300 | 90.4 ± 0.09 | 90.0 ± 0.11 | 90.0 ± 0.15 | 90.1 ± 0.03 |
| Mean corpuscular hemoglobin (MCH) | T0 | 29.9 ± 0.11 | 30 ± 0.09 | 30.0 ± 0.19 | 29.9 ± 0.03 |
| | T3 | 30.0 ± 0.15 | 29.9 ± 0.14 | 30.0 ± 0.05 | 30.0 ± 0.09 |
| | T30 | 30.0 ± 0.07 | 30.1 ± 0.11 | 30.1 ± 0.06 | 29.9 ± 0.08 |
| | T300 | 30.1 ± 0.06 | 30 ± 0.01 | 30 ± 0.09 | 30.0 ± 0.06 |
| Mean corpuscular hemoglobin concentration (MCHC) | T0 | 33.3 ± 0.13 | 33.3 ± 0.09 | 33.3 ± 0.14 | 33.2 ± 0.16 |
| | T3 | 33.3 ± 0.18 | 33.2 ± 0.07 | 33.3 ± 0.12 | 33.3 ± 0.17 |
| | T30 | 33.3 ± 0.14 | 33.4 ± 0.17 | 33.4 ± 0.09 | 33.2 ± 0.03 |
| | T300 | 33.4 ± 0.12 | 33.3 ± 0.12 | 33.3 ± 0.07 | 33.3 ± 0.09 |

T0, T3, T30, and T300 correspond to 0 ng EE2 plus 0 ng DES, 3 ng EE2 plus 15 ng DES, 30 ng EE2 plus 150 ng DES, and 300 ng EE2 plus 1500 ng DES per liter of water fishes were exposed to, respectively. Values with different alphabetical superscripts in a row differ significantly ($p < 0.05$) among different treatments. All values are expressed as mean ± SD.

### 3.5. Erythrocytic Cellular Abnormalities (ECA)

A number of erythrocytic cellular abnormalities, such as twin, fusion, teardrop, echinocytes, and elongated shaped were observed in the blood smears of fish treated with different EE2/DES mixtures (Figure 3). In general, significantly higher frequencies of these ECAs were observed in all the treatments compared to the control. For both sampling days, the frequencies of ECAs were significantly higher in fish reared in T300 and T30 compared to T3 and T0 (Table 5).

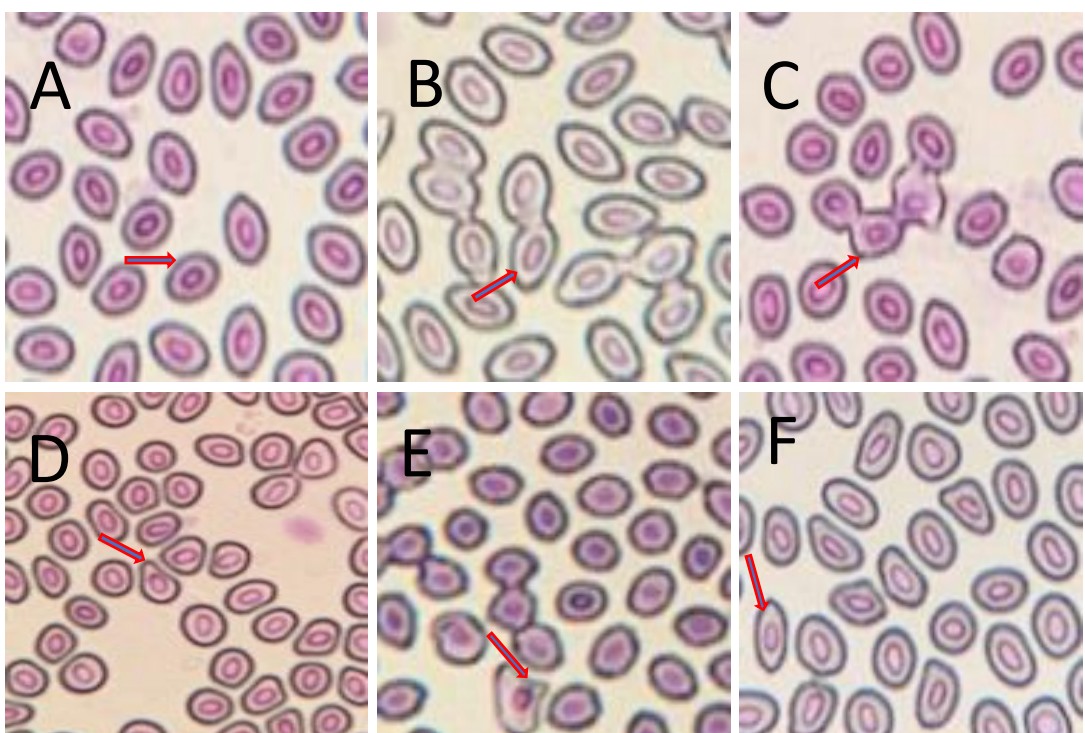

**Figure 3.** Various erythrocytic cellular abnormalities (ECA) in Giemsa-stained blood smears of climbing perch exposed to different concentrations of ethinylestradiol (EE2)/desogestrel (DES) mixture. Arrows indicate different abnormalities—(**A**) regular (no abnormality), (**B**) twin, (**C**) fusion, (**D**) teardrop-shaped, (**E**) echinocytic, and (**F**) elongated erythrocytes.

**Table 5.** Frequencies of erythrocytic cellular abnormalities (ECA) in climbing perch exposed to different concentrations of ethinylestradiol/desogestrel mixtures.

| ECA | Treatments | Percentage of ECA | | | |
|---|---|---|---|---|---|
| | | Male | | Female | |
| | | 45 Day | 60 Day | 45 Day | 60 Day |
| Teardrop | T0 | $0.21 \pm 0.01$ [a] | $0.51 \pm 0.01$ [a] | $0.12 \pm 0.01$ [a] | $0.22 \pm 0.01$ [a] |
| | T3 | $0.63 \pm 0.04$ [b] | $0.93 \pm 0.05$ [b] | $0.55 \pm 0.07$ [b] | $0.45 \pm 0.05$ [b] |
| | T30 | $1.15 \pm 0.04$ [c] | $1.25 \pm 0.04$ [c] | $0.55 \pm 0.04$ [b] | $0.59 \pm 0.04$ [b] |
| | T300 | $1.63 \pm 0.11$ [c] | $1.73 \pm 0.17$ [c] | $0.83 \pm 0.09$ [c] | $0.98 \pm 0.07$ [c] |
| Fusion | T0 | $0.38 \pm 0.03$ [a] | $0.35 \pm 0.03$ [a] | $0.36 \pm 0.05$ [a] | $0.39 \pm 0.05$ [a] |
| | T3 | $0.47 \pm 0.05$ [b] | $0.75 \pm 0.09$ [b] | $0.61 \pm 0.07$ [b] | $0.64 \pm 0.05$ [b] |
| | T30 | $0.87 \pm 0.06$ [c] | $0.79 \pm 0.06$ [b] | $1.11 \pm 0.09$ [c] | $0.91 \pm 0.09$ [b] |
| | T300 | $0.98 \pm 0.09$ [c] | $0.89 \pm 0.11$ [c] | $1.63 \pm 0.07$ [c] | $1.85 \pm 0.09$ [c] |
| Elongated | T0 | $0.68 \pm 0.03$ [a] | $0.81 \pm 0.03$ [a] | $0.86 \pm 0.05$ [a] | $0.80 \pm 0.03$ [a] |
| | T3 | $1.11 \pm 0.04$ [b] | $1.22 \pm 0.07$ [a] | $1.01 \pm 0.07$ [b] | $1.11 \pm 0.05$ [a] |
| | T30 | $1.47 \pm 0.06$ [c] | $1.56 \pm 0.09$ [b] | $1.11 \pm 0.11$ [b] | $1.14 \pm 0.07$ [b] |
| | T300 | $1.55 \pm 0.07$ [c] | $1.72 \pm 0.11$ [c] | $1.53 \pm 0.01$ [c] | $1.18 \pm 0.03$ [b] |
| Echinocytic | T0 | $0.11 \pm 0.01$ [a] | $0.17 \pm 0.03$ [a] | $0.21 \pm 0.01$ [a] | $0.41 \pm 0.01$ [a] |
| | T3 | $0.43 \pm 0.07$ [a] | $0.65 \pm 0.05$ [b] | $0.51 \pm 0.03$ [b] | $0.73 \pm 0.03$ [b] |
| | T30 | $0.95 \pm 0.11$ [b] | $1.15 \pm 0.19$ [c] | $0.89 \pm 0.05$ [b] | $0.98 \pm 0.05$ [b] |
| | T300 | $1.43 \pm 0.18$ [b] | $1.73 \pm 0.28$ [c] | $1.53 \pm 0.17$ [c] | $1.89 \pm 0.19$ [c] |

**Table 5.** *Cont.*

| ECA | Treatments | Percentage of ECA | | | |
|---|---|---|---|---|---|
| | | Male | | Female | |
| | | 45 Day | 60 Day | 45 Day | 60 Day |
| Twin | T0 | 0.31 ± 0.01 [a] | 0.91 ± 0.01 [a] | 0.12 ± 0.01 [a] | 0.22 ± 0.01 [a] |
| | T3 | 0.63 ± 0.04 [b] | 1.13 ± 0.05 [b] | 0.65 ± 0.07 [b] | 0.45 ± 0.05 [b] |
| | T30 | 1.05 ± 0.04 [b] | 1.45 ± 0.04 [c] | 0.45 ± 0.04 [b] | 0.55 ± 0.04 [b] |
| | T300 | 1.93 ± 0.11 [c] | 1.98 ± 0.17 [c] | 0.93 ± 0.09 [c] | 0.83 ± 0.07 [c] |

T0, T3, T30, and T300 correspond to 0 ng EE2 plus 0 ng DES, 3 ng EE2 plus 15 ng DES, 30 ng EE2 plus 150 ng DES, and 300 ng EE2 plus 1500 ng DES per liter of water fishes were exposed to, respectively. Values with different alphabetical superscripts in a row differ significantly ($p < 0.05$) among different treatments. All values are expressed as mean ± SD.

## 4. Discussion

### 4.1. Alterations in GSI and HSI Values

In the current study, although there was no definite pattern in male GSI values, the same for females was significantly higher with increasing concentrations of EE2/DES mixture. Significantly higher GSI values in females may have been caused by the excessive metabolic activity of the liver in metabolizing the chemical and may have produced more lipids and proteins compared to the normal conditions, which might have been stored in the gonads, enlarging the size of the ovary and thus resulting in an increase in GSI values [65]. Belt et al. (2009) reported that adult zebrafish exposed to EE2 showed a decrease in GSI values, which contradicts our result [66]. However, Zha et al. (2007) observed that adult minnow (*Gobiocypris rarus*) after exposure to ethinylestradiol and nonylphenol had increased GSI values, which is similar to our research [67].

In this study, although HSI values in females did not differ between control and treatments, treated fishes had significantly higher HSI values in the case of males. HSI values of female rare minnow (*Gobiocypris rarus*) exposed to three EE2 concentrations were lower than controls [67]. In contrast, Parrott et al. (2005) reported that fathead minnows (*Pimephales promelas*) exposed to high EE2 concentrations showed higher HSI values, similar to our findings for male climbing perch [68].

### 4.2. Histolopathogical Alterations in Ovary

The hypothalamic–pituitary–gonadal axis intensively maintains the reproduction of bony fishes and most vertebrates and depends on the response mechanism of steroid hormones such as estrogens that play a significant role in successful reproduction. Estrogens produced in the ovary may affect the hypothalamus, pituitary, and gonads either positively or negatively [69]. The current research findings showed different disorders in the ovarian histopathology of climbing perch exposed to different mixtures of EE2/DES. Degenerated granulose layers, atretic follicles, interfollicular space, adhesion, cytoplasmic clumping, cysts, degenerated oocyte walls, and necrosis in the ovarian tissues were observed in climbing perch. Kang et al. (2008) reported that medaka (*Oryzias latipes*) exposed to different concentrations of methyl testosterone showed degeneration of oocytes, atretic follicles, and degenerated ovarian follicles in all the treatments, which is similar to our research result [70]. Zebrafish exposed to low concentrations of ethinylestradiol (10 ng/L and 25 ng/L) showed a reduction in the mature yolk-filled oocytes. Zha et al. (2007) reported that adult rare minnow (*Gobiocypris rarus*) exposed to low EE2 concentrations (25 ng/L) also showed degeneration in ovarian tissues, which is similar to our findings [67].

### 4.3. Histological Alterations in Testes

A number of testicular cell deformities were observed in climbing perch exposed to three different concentrations of EE2/DES mixture, including increased interstitial area, ruptured tubular epithelia, the prevalence of cysts, and degenerated or absent Sertoli cells. A similar result was reported by Zha et al. (2007) for minnow (*Gobiocypris rarus*) exposed to

5 ng/L and 25 ng/L EE2, which were found to have primary and secondary spermatogonia but no sperms, whereas the seminiferous tubules of the control fish were filled with fertile sperms.

Tench (*Tinca tinca*) exposed to sub-lethal doses of EE2 (50, 100, and 500 μg/Kg t.w.) for 30 days were found to have losses of the normal tubular structure, a decrease in the number of tubules, and degeneration in Sertoli and Leydig cells associated with an increase in necrotic testicular cells as well as primary oocytes in the testis at very high levels of EE2 (100 and 500 μg/Kg t.w.) [71]. Rainbow trout (*Oncorhynchus mykiss*) male fry (136 days post-hatch) exposed to EE2 1 μg/L were found with several degenerations of the testis such as loss of tubular arrangement, loss of germ cells, differentiation, and presence of lacuna [72]. Similar to our findings, Miller et al. (2012) reported that breeding medaka fish exposed to ethinylestradiol (EE2) at 1 μg/L and 10 μg/L showed depleted germ cell epithelia, thickening of the interstitial area, testicular oocytes, and karyomegalic germ cells in some testis samples [73].

### 4.4. Alterations of Blood Cells and Glucose Levels

In the current research, blood glucose levels were found to be decreased significantly with increasing chemical (EE2/DES) concentrations and exposure period for both male and female fish. Kavitha et al. (2010) reported that Indian major carp *Catla catla* were exposed to sublethal concentrations of arsenate and showed a significant decrease in glucose levels compared to the control group, which is similar to our research findings [74]. This reduction in glucose levels may be due to hypoxic conditions caused by the stressors which lead to an excess utilization of stored carbohydrates [75]. However, some experiments also reported that glucose levels increased for the freshwater teleost fish *Cyprinus carpio* exposed to lindane, and the neotropical fish *Prochilodus lineatus* exposed to diesel oil [76,77]. This increase may be attributed to stressed fishes' increased response to gluconeogenesis in order to satisfy their increased energy requirements [78]. Hemoglobin and hematocrit (PCV) levels were observed to be higher in chemical (EE2/DES)-treated groups compared to the control group whereas no changes were found in MCV, MCH, and MCHC values among the treatments and control groups in both male and female fish throughout the experimental period. A similar result was reported by Svoboda et al. (2001), who found a significant decrease in hemoglobin and hematocrit levels compared to the control group in common carp (*Cyprinus carpio* L.) exposed to diazinon where MCV, MCH, and MCHC values were comparable in both groups [79]. Indian major carp *Catla catla* exposed to arsenate showed reduced hemoglobin and hematocrit (PCV) levels compared to the control group but MCV, MCH, and MCHC were found to be increased in treatments compared to the control [77]. The decrease in hemoglobin levels in climbing perch reported in this study may be due to the chemicals' destructive effect on erythropoietic tissue, affecting the cells' viability. RBC counts for both male and female fish was observed to be lower in the higher treatment groups compared to the control group. Talas et al. (2010) reported that rainbow trout (*Oncorhynchus mykiss*) exposed to different concentrations of propolis had a significant decrease in RBCs in the treatment groups compared to the control group, which is similar to our findings [80]. The decline in RBC count is caused by erythropoiesis inhibition and an increase in the rate of erythrocyte degradation in hematopoietic organs [81]. Thus, a reduction in the number of red blood cells means a reduction in oxygen and carbon dioxide transport as well as alterations in blood flow in climbing perch, which may cause anemia [82].

In contrast to RBCs, WBCs were found to be increased in chemical (EE2/DES)-treated groups compared to the control group. Rainbow trout exposed to propolis had an increased leukocyte count (WBCs) in the treatment compared to the control [83], which is supportive of the current findings. Indian major carp *Catla catla* exposed to different concentrations of arsenate had a decrease in WBCs in treatments compared to controls, which is opposite to our result [77]. In many species, white blood cells play a role in the control of immune function, and an increase in WBCs in stressed animals suggests a defensive response to

stress. This increase in WBCs may be due to leucocytosis under chemical stress, deemed an adaptive value for the tissue.

### 4.5. Alterations in Erythrocytic Cellular Structure

In the present study, we have observed various morphological (cellular and nuclear) alterations in the blood cells of fish exposed to chemical mixtures. Blood smears of fish treated with chemical mixtures result in ECAs. The frequencies of ECA were higher in the chemically treated groups compared to the control group. Similarly, it was reported that Nile tilapia (*Oreochromis niloticus*) exposed to textile industry effluents had many nuclear and cytoplasmic deformities in blood cells [84].

## 5. Conclusions

The current research sheds new light on the effects on climbing perch of an oral contraceptive pill containing a synthetic estrogen mixture of ethinylestradiol (EE2) and desogestrel (DES). Although there was no discernible patterns in male GSI values in the current study, female GSI values increased significantly as the EE2/DES mixture concentration increased. Increased GSI values in females may have resulted from the liver's excessive metabolic activity in metabolism, which may have produced more lipids and proteins than normal conditions, which may have been stored in the gonads causing ovary enlargement, resulting in an increase in GSI values. The current study discovered degenerated granulose layers, atretic follicles, interfollicular space, adhesion, cytoplasmic clumping, cysts, degenerated oocyte walls, and necrosis in the ovarian tissues of females exposed synthetic estrogen. Similarly, a number of testicular cell deformities in male fish were observed, including increased interstitial area, ruptured tubular epithelia, cysts, and degenerated or absent Sertoli cells. These finding suggest that exposure to synthetic estrogen could cause significant impacts on reproductive performance in both sexes. Blood glucose levels were found to be significantly lower with increasing chemical (EE2/DES) concentrations and exposure periods in both male and female fish, which might be due to estrogen-dependent depression of gluconeogenesis. Lower blood glucose levels in rainbow trout and whitefish were also observed, which could be mediated by a reduction in glycogen deposition [60,85,86]. We detected morphological (cellular and nuclear) alterations in the blood cells of fish subjected to the chemical combination. ECAs were found in blood smears of fish treated with the chemical combination. ECAs were more common in the chemically treated groups than in the control group. The impact of estrogen exposure is very well investigated in various species [87–91].

Fish such as climbing perch are a significant dietary source of environmental contaminants with endocrine action in humans, especially when they constitute a significant portion of food consumption. Many chemicals exist in trace amounts in the environment but have significant biological activity. Estimating biota health hazards and degrees of exposure to environmental contaminants is crucial. The scientific evidence demonstrates a link between animals' reproductive health and chronic exposure to low levels of contaminants in the environment or food chain. Subtle health effects have been recorded in certain Arctic populations exposed to a variety of contaminants prevalent in the food chain (in traditional foods), including endocrine-disrupting substances. The possibility that the bio-accumulative properties of persistent organic chemicals with hormone-like activity, as well as chronic low-level exposure, may contribute to overall breast cancer risk in women, as well as reproductive and developmental effects in humans, has significant implications for the prevention of these diseases in Western countries. As discovered in our work, as well as the emerging literature, estrogen exposure poses a considerable harm to fish populations, with male fish exposed to estrogen shown to become feminine and a significant rise in female progeny [87–91]. There is published evidence linking estrogens in the environment to breast cancer. However, there are significant gaps in our understanding of estrogen levels in the environment, necessitating a global effort to collect more data from a larger number of sample locations. The synthetic estrogen ethinylestradiol is more

persistent in the environment than natural estrogens and may be a greater environmental issue. Overall, the effects of estrogen on people and aquatic organisms, as well as various tactics and future directions, were meticulously evaluated in Adeel et al. [89].

**Author Contributions:** C.W.: Methodology; writing original draft. N.A.: Methodology; writing original draft. M.H.U.: Data curation; Methodology. M.R.: reviewing. K.A.S.: Data curation; formal analysis. M.J.I.: reviewing; editing. R.K.B.: Investigation; validation; visualization. H.R.: Conceptualization; supervision; writing; reviewing; editing. All authors have read and agreed to the published version of the manuscript.

**Funding:** This research received no external funding.

**Institutional Review Board Statement:** The experimental animal protocol was reviewed and approved by the Asian University of Women's institutional review board and ethics review committee.

**Informed Consent Statement:** Not applicable.

**Data Availability Statement:** The data presented in this study are available on request from the corresponding author.

**Conflicts of Interest:** The authors declare no competing interests.

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
