# Peer review of "Contraceptive-Pill-Sourced Synthetic Estrogen and Progestogen in Water Causes Decrease in GSI and HSI and Alters Blood Glucose Levels in Climbing Perch (Anabas testudineus)"

_2673-9917, doi:10.3390/hydrobiology2010002_

Round 1
Author Response
We are very glad for your comments, please find our responses in attached word document.
Best regards,

Reviewer 2 Report
This study evaluated the changes in gonads and hematological parameters in climbing perch (Anabas testudineus) exposed to synthetic estrogen. The residues of pharmacological substances used in the veterinary and human fields can contaminate the aquatic environment and interfere with the physiology of aquatic organisms. This is an important aspect, especially when it comes to hormonal substances whose biological effects are significant. Exposure to estrogens can have various detrimental effects in fish. It can reduce general viability, induce gonadal malformations or feminization of genetic males, or lead to sterilization. Changes in blood parameters represent the rapid response when the fish are exposed to synthetic estrogen.
The introduction is relevant but it necessary to improve it and add new references.
Rewrite the table it’s not well organized …..see below comment
The discussion, in the light of the obtained results and of knowledge should be explain better.
According to my opinion, the manuscript can be accepted for publication after major revision.
Below some corrections are reported.
Title: change it because only glucose was analysed as biochemical parameters so it this might confuse the readers...
Introduction:
Please add in line 82-84 this sentence “Alterations in blood parameters associated to environmental pollutants and represent an important tool in order to assessing the health of fish (Bath et al., 2022, Fazio F., 2019) “ add these references:
Bhat RA, et al. Effects of heavy pollution in different water bodies on male rainbow trout (Oncorhynchus mykiss) reproductive health. Environ Sci Pollut Res Int. 2022 Nov 2. doi: 10.1007/s11356-022-23670-w. Epub ahead of print. Erratum in: Environ Sci Pollut Res Int. 2022 Nov 14;: PMID: 36322349.
Fazio F., Fish hematology analysis as an important tool of aquaculture: A review,
Aquaculture, Volume 500, 2019, Pages 237-242
Add this important review:
Muhammad Adeel, Xiaoming Song, Yuanyuan Wang, Dennis Francis, Yuesuo Yang, Environmental impact of estrogens on human, animal and plant life: A critical review, Environment International,
Volume 99, 2017, Pages 107-119,
Lines 86 -93 move to the beginning of the introduction. It is necessary to conclude the part of the introduction by specifying what the purpose of the research is and this is not indicated. Please rewrite this part.
2. Materials and Methods
2.1 Experimental fish
The experimental design of this manuscript needs some clarification:
Please indicate the number of fish used in trial
The authors state “Fish handling and experimental protocols were approved by the Institutional 104 Review Board of the Asian University for Women” insert number authorization and ethic document
2.2 Experimental setup and chemical exposure
The authors write “…tank and acclimatized at laboratory conditions for 21 days prior to starting the trial…” Indicate if mortality were observed during the acclimation and trial. Indicate the photoperiod L/D and time of feed.
How was the concentration of administered estrogen chosen? there are references in the literature
Indicate the Water quality parameters assessment…..
Indicate the weight and length fork before the trial and after trial (gain weight )
Insert information for blood sample storage and time before haematological analysis.
For the hematological parameters obtained, have these been compared with values obtained for this particular species in previous works?
The experimental part of tissue analysis and blood sampling is very confusing. Were the fish anesthetized before collection and then sacrificed???? explain better
Statistical:
The statistical analysis used is appropriate.
3.4 Changes in hemato-biochemical parameters
Change title only glucose as biochemical parameters …
Discussion
4.4 Alterations of hemato-biochemical parameters
this part is not very convincing the authors report that the changes in blood glucose are caused by stress and those of the haematological parameters depend on causes related to oxygen. But what is the relationship between the estrogen administered and the blood values studied? it is necessary to explain this concept
5. Conclusion
what is the impact of these estrogens on human health? is it possible to have an endocrine risk for humans who eat fish with estrogen? is there a limit value? what directions in the future??
Author Response

(The authors gave the same response as above.)

Round 2
Reviewer 2 Report
The authors answered all queries and the manuscript was improved. My decision is accepted.
Author Response
We have incorporated the comments and updated the manuscript.